

# Trajectories of change after a health-education program in Japan: decay of impact in anxiety, depression, and patient-physician communication

Min Jeong Park[1,2], Joseph Green[3], Hun Sik Jung[4] and Yoon Soo Park[5]

[1] Department of Nursing, College of Nursing, Konyang University, Daejeon, South Korea
[2] Department of Health Communication, University of Tokyo, Tokyo, Japan
[3] Graduate School of Medicine, University of Tokyo, Tokyo, Japan
[4] College of Global Business, Konyang University, Nonsan, South Korea
[5] College of Medicine, University of Illinois at Chicago, Chicago, IL, USA

Corresponding author
Joseph Green, jgreen-tky@umin.ac.jp

## ABSTRACT

**Background:** Health education can benefit people with chronic diseases. However, in previous research those benefits were small, and reinforcement to maintain them was not effective. A possible explanation is that the benefits *appeared* to be small and reinforcement *appeared* to be ineffective because those analyses mixed data from two latent groups: one group of people who needed reinforcement and one group of people who did not. The hypothesis is that mixing the data from those two different groups caused the true effects to be "diluted."

**Methods:** To test that hypothesis we used data from the Chronic Disease Self-Management Program in Japan, focusing on anxiety, depression, and patient-physician communication. To identify latent trajectories of change after the program, we used growth-mixture modeling. Then, to find out which baseline factors were associated with trajectory-group membership, we used logistic regression.

**Results:** Growth-mixture modeling revealed two trajectories—two groups that were defined by distinct patterns of change after the program. One of those patterns was improvement followed by backsliding: decay of impact. On anxiety and depression the decay of impact was large enough to be clinically important, and its prevalence was as high as 50%. Next, logistic regression analysis revealed that being in the decay-of-impact group could be predicted from multimorbidity, low self-efficacy, and high scores on anxiety or depression at baseline. In addition, one unexpected finding was an association between multimorbidity and *better* patient-physician communication.

**Conclusions:** These results support the hypothesis that previous findings (i.e., *apparently* small effect sizes and *apparently* ineffective reinforcement) actually reflect "dilution" of large effects, which was caused by mixing of data from distinct groups. Specifically, there was one group with decay of impact and one without. Thus, evaluations of health education should include analyses of trajectory-defined groups. These results show how the group of people who are most likely to need reinforcement can be identified even before the educational program begins. Extra attention and reinforcement can then be tailored. They can be focused specifically to benefit the people with the greatest need.

## INTRODUCTION

Nothing lasts forever, yet a worthy goal of health education is for its benefits to be sustained. In some studies, benefits of health education have been found to last longer than 6 months (*Barlow et al., 2005*, *2008*; *Brady et al., 2013*), while in others the findings are more nuanced—some improvements do not endure (*Caplin & Creer, 2001*; *Clark, 2003*; *Franks et al., 2009*; *Hennessy et al., 1999*; *Krebs, Prochaska & Rossi, 2010*; *Lorig et al., 2004*; *Norris et al., 2002*). Particularly with regard to the self-management educational interventions that are used to help people reduce the impact of chronic diseases, it was recently noted that those interventions might need improvements to ensure that they have longer-lasting impact (*Miller et al., 2015*), which at least implicitly acknowledges that their immediate benefits can decay after their (short-lived) demonstrated period of effectiveness. To sustain these programs' benefits, reinforcement has been recommended (*Clark, 2003*; *Green, 1977*; *Newman, Steed & Mulligan, 2009*), but reinforcement has generally not been found to be useful (*Glasgow et al., 2002*; *Lorig & Holman, 1989*; *Lorig et al., 2006*; *Lorig et al., 2008*; *Nguyen et al., 2005*; *Riemsma, Taal & Rasker, 2003*).

In this context, we consider the Chronic Disease Self-Management Program (CDSMP), which can improve health status and can increase the frequency of desirable health-related behaviors (*Foster et al., 2007*; *Franek, 2013*; *Whitelaw et al., 2013*). While the benefits of the CDSMP are "statistically significant," some of them have also been described as minimal (*Franek, 2013*) or moderate (*Brady et al., 2013*). The program's developers attributed these "modest" effect sizes to heterogeneity in the clinical and demographic characteristics of the participants (*Lorig et al., 2006*), which points to a need to understand differences among participants and factors that might contribute to larger and more-sustained benefits.

In general, treating participants as homogeneous conceals true heterogeneity. For example, an intervention may be very useful in some participants, but that fact will not be recognized if one examines only the average for the group as a whole (*Moynihan, Henry & Moons, 2014*). In addition, important heterogeneity in treatment effects can occur not only across groups but also over time. Sustained improvement in some participants could obscure relapse in others. Reinforcement given to all can appear to be ineffective, even if it is quite useful to some.

In previous research, subsets of participants were defined by socio-demographic characteristics, personality factors, etc., and not by patterns of change after the intervention (*Franks et al., 2009*; *Harrison et al., 2012*; *Jerant et al., 2010*; *Reeves et al., 2008*; *Swerissen et al., 2006*; *Smeulders et al., 2010*). In contrast, *Green (1977)* discussed five distinct patterns (i.e., trajectories) of change after health education. He referred to the pattern in which good outcomes do not endure as *decay of impact* (also sometimes called relapse or backsliding). Its essence is simple: deterioration after improvement. Outside of health

education, a similar concept is used commonly in research on the effectiveness of treatments for addiction (*Hendershot et al., 2011*; *Menon & Kandasamy, 2018*). Within the field of health education, *Green (1977)* may have been the first to point out that decay of impact can cause the benefits of an intervention to *appear* to be larger or smaller than they really are, depending on when the outcome is measured. That difference between the apparent change and the real change can be minimized only by measuring the outcome repeatedly over a relatively long period of time. *Hennessy et al. (1999)* measured self-efficacy for condom use five times during 1 year after an HIV-prevention educational program, and they found clear evidence of decay of impact 3 months after enrollment in the program. *Caplin & Creer (2001)* identified various characteristics of people with decay of impact 7 years after an asthma self-management program: For example, the people with decay of impact had initially been less *self*-motivated to participate in the program. Nonetheless, with only a few exceptions, over more than 40 years since *Green (1977)* described decay of impact in health education, that concept seems to have received little attention from researchers. Stated simply, decay of impact as a person's pattern of change after chronic-disease self-management education has been studied only very rarely. But knowing the magnitude of decay, and knowing when and in whom it occurs would be very useful in evaluating a health-education program's effectiveness, and also in targeting interventions. Information about decay of impact can give planners an objective basis for deciding whether reinforcement is needed, when it is needed (*Hennessy et al., 1999*), and who is most likely to need it (*Park et al., 2012*).

Taking seriously the possibility of decay of impact entails defining groups by their patterns of change, that is, the trajectories of their outcomes during follow-up. Using data collected in two waves, *Nolte et al. (2007)* may have been the first to study groups defined by their change after health education. An extension from patterns defined using two waves of data to those defined using four was published a few years later (*Park et al., 2013*). However, without the use of a well-established method for analyzing longitudinal data, doubts remain regarding different trajectories in outcomes. In this current study, using data collected before and after the CDSMP we applied growth-mixture modeling (GMM) to empirically identify latent groups that were characterized by their patterns of longitudinal change, and we subsequently used logistic regression to identify baseline factors contributing to group membership (*Cook et al., 2015*; *Hibbard et al., 2007*; *Muthén, 2004*; *Rabe-Hesketh, Skrondal & Pickles, 2004*; *Ram & Grimm, 2009*). This approach allows more granularity in evaluating the program's effects and more accuracy in assessing needs for reinforcement.

Many outcomes of the CDSMP have been studied (*Foster et al., 2007*; *Lorig et al., 1996*), but here we focused only on symptoms of anxiety, symptoms of depression, and the use of proactive techniques to improve patient-physician communication. Thus, the focus is not on the longitudinal change in participants' chronic illnesses themselves (with the possible exception of the small number of participants who had clinical depression). Trajectories of change in anxiety and depression could reflect changes in the application of coping skills learned in the CDSMP, while patient-physician communication is of course a health-related behavior. We chose those three outcomes because all of them are

relevant in many different chronic medical conditions, and thus the results will be important to a large proportion of all CDSMP participants. Also, one area on which the CDSMP focuses specifically is learning and practicing better patient-physician communication.

## METHODS

### Participants

In this study, we focused on a particular population: adults in Japan who have at least one chronic medical condition and who participated in the CDSMP. In this implementation of the CDSMP in Japan, the program accommodated people who were community-dwelling, were able to attend weekly group meetings, and were able to communicate easily in Japanese. As in all implementations of the CDSMP, the participants had various different chronic medical conditions. The population of CDSMP participants in Japan consists only of adults (defined as people who were at least 18-years-old), but it is otherwise diverse with respect to age, duration of chronic illness, and other socio-demographic and clinical characteristics (Appendix 1).

Chronic Disease Self-Management Program participants were recruited through public service centers, outpatient clinics of hospitals, and the Internet homepage of the *Japan Chronic Disease Self-Management Association (2018)*. Some participants also found out about the program through friends or acquaintances. Participation was voluntary. All people who met the conditions described above comprised the population of interest. The study itself involved a census (not a sample) of that population.

### The program

Based on self-efficacy theory (*Bandura, 2019*), the program aims to build the participants' skills in six areas: (1) handling pain, fatigue, frustration, and isolation, (2) exercising to maintain and increase strength, endurance, and flexibility, (3) using medications appropriately, (4) improving communication with friends, family, and healthcare professionals, (5) achieving and maintaining proper nutrition, and (6) evaluating new therapies (*Self Management Resource Center, 2018*). Skills in those areas were taught and practiced during group-discussion sessions that were held once each week for 6 weeks. Each group had two lay facilitators who had undergone approximately 35 h of training. A textbook was used as the reference for the program's content (*Lorig et al., 2001*).

### Measures

Demographic and clinical information were collected using self-administered questionnaires. Also included in the questionnaires were scales to measure health status, health-related behaviors, psychological variables, etc. Those included self-efficacy to manage chronic health conditions (on a 0-to-60 scale, coefficient alpha = 0.92) and the hospital anxiety and depression scale (HADS, *Matsudaira et al., 2009*). The HADS asks about symptoms of anxiety and of depression in the past week. Possible total scores on both the anxiety scale and the depression scale range from 0 to 21, with higher scores reflecting more symptoms and more frequent symptoms. Coefficient alpha for the

depression scale of the HADS was 0.73 and for the anxiety scale it was 0.84. Also included was a 3-item scale to measure communication with physicians, with possible total scores ranging from 0 to 15 (coefficient alpha = 0.78). Higher scores reflected more frequent use of proactive methods for good patient-physician communication (*Lorig et al., 1996*).

## Study design and timing of measurements

Data were collected four times over 1 year. Baseline data were collected before the first group-discussion session. Follow-up questionnaires were sent by postal mail 3, 6, and 12 months later. A post-paid envelope addressed to the research office was included for returning the questionnaire. To ensure that data collection was as complete as possible, one of the researchers (MJP) was available to speak directly with participants and answer their questions about the study. In addition, a reminder postcard was sent whenever a follow-up questionnaire was not received by return mail within 2 weeks.

## Analyses

To allow detection of decay of impact, the analyses were done using data from participants who provided at least three waves of data (456/643; 71%). Unconditional quadratic growth curves (*Rabe-Hesketh, Skrondal & Pickles, 2004*) and conditional growth curves were fit using the four-wave data with all participants. Time-variant (self-efficacy) and time-invariant covariates (gender, educational status, partnered status, number of diagnoses, and history of illness) were included in the growth-curve analysis, with anxiety, depression, and communication as outcomes, to examine baseline differences and interactions over time. Quadratic terms were specified on "time" by including a squared time (i.e., time × time) term. Interaction terms between time-invariant covariates and time were included to examine factors contributing to longitudinal changes.

Growth-mixture modeling (GMM) was used to fit quadratic growth curves for 2, 3, and 4 latent groups. Posterior mode estimation, a partly Bayesian approach, was used to obtain consistent parameter estimates and to avoid boundary estimation issues (*Park, Xing & Lee, 2018*; *Huang & Bandeen-Roche, 2004*). Moreover, to obtain optimal estimation, we began with 100 sets of starting values and used a combination of expectation-maximization algorithm followed by the Newton–Raphson method to obtain parameter estimates and model fit indices. Finally, we examined the Jacobian matrix to be of full rank to ensure local identification of results. This process was repeated for each growth-mixture model allowing for the different latent groups. We decided on the final number of groups after examining the relative fit (Bayesian information criterion (BIC)) and absolute fit (proportion correctly classified ($P_c$) based on posterior probability). Multiple logistic regression was used to identify factors contributing to trajectory-group membership. Of the 456 participants whose data were analyzed by GMM here, 369 were included in a previous non-GMM analysis (*Park et al., 2013*). Data were analyzed with Latent Gold 5.1 (Belmont, MA, USA), Stata 14 (College Station, TX, USA), and JASP (https://jasp-stats.org/).

## Ethics

This study was approved by the University of Tokyo (number 1472–(2), Research Ethics Committee, Graduate School of Medicine). Participation in the CDSMP and in this

**Table 1 Descriptive statistics for all CDSMP participants considered together, baseline and follow up over 1 year.**

| Factor | Baseline | | | 3 Months | | | 6 Months | | | 12 Months | | |
|---|---|---|---|---|---|---|---|---|---|---|---|---|
| | *n* | Mean | SD | *n* | Mean | SD | *n* | Mean | SD | *n* | Mean | SD |
| Self-efficacy: 0–60, higher scores are better | 456 | 32.33 | 12.54 | 415 | 34.48 | 12.02 | 425 | 35.16 | 12.10 | 404 | 35.14 | 12.99 |
| Anxiety: 0–21, lower scores are better | 456 | 6.89 | 4.25 | 423 | 6.27 | 4.00 | 428 | 6.03 | 4.22 | 406 | 6.43 | 4.62 |
| Depression: 0–21, lower scores are better | 456 | 7.21 | 3.82 | 423 | 6.69 | 3.64 | 428 | 6.53 | 3.93 | 406 | 6.77 | 4.20 |
| Communication: 0–15, higher scores are better | 456 | 6.22 | 3.77 | 422 | 6.59 | 3.99 | 426 | 6.91 | 4.11 | 401 | 6.77 | 4.18 |

Note:
The outcomes discussed here are anxiety, depression, and communication with physicians. Self-efficacy was used as a mediator in subsequent analyses because of its importance in the theoretical basis of the CDSMP.

research were voluntary. Informed consent was obtained in writing from all participants before the study began.

# RESULTS

## The participants

Data from 456 participants were analyzed. Among them, 79% were women, 48% were college educated, 52% were partnered (married or living with someone), and 47% had more than one chronic condition. Details of multimorbidity are in Appendix 1.

## All participants considered together

In the analysis with all participants considered together (Table 1), for the first 6 months communication with physicians increased, while both anxiety and depression decreased. However, by the end of the follow-up year all three outcomes had begun changing back toward their baseline values. That is, the initial improvements appeared to be followed by at least some backsliding.

Similarly, the growth-curve analysis also indicated that change over time was curvilinear for all three outcomes: *p*-values for the quadratic terms for anxiety, depression, and communication were 0.001, 0.038, and 0.002, respectively (Table 2). In addition, higher self-efficacy at baseline was associated with less anxiety, less depression, and better communication with physicians.

Also associated with both anxiety and communication was the number of diagnoses, and the regression coefficients for both were positive (Table 2). That is, the participants with more comorbid conditions had greater anxiety at baseline. It is noteworthy that the participants with more comorbid conditions also had *better* baseline scores on the scale measuring communication with physicians. These associations did not change over time, as evidenced by the small coefficients for the terms representing interaction with time.

## Groups defined by their trajectories

For all three outcomes, the GMM results were similar: The BIC and the $P_c$ both led to the conclusion that the best-fitting models were those with two groups (Appendix 2).

For each outcome, those two groups began from substantially different baseline scores (Fig. 1). Also for each outcome, one group changed very little throughout the follow-up

**Table 2 Results of growth-curve analysis, all CDSMP participants considered together ($n = 456$).**

| Factor | Anxiety | | | Depression | | | Communication | | |
|---|---|---|---|---|---|---|---|---|---|
| | Coefficient | Std err | p-value | Coefficient | Std err | p-value | Coefficient | Std err | p-value |
| Fixed effect | | | | | | | | | |
| Time | −0.24 | (0.07) | 0.001 | −0.14 | (0.07) | 0.034 | 0.20 | (0.06) | 0.001 |
| Time × Time | 0.01 | (0.00) | 0.001 | 0.01 | (0.00) | 0.038 | −0.01 | (0.00) | 0.002 |
| Self-efficacy | −0.10 | (0.01) | <0.001 | −0.11 | (0.01) | <0.001 | 0.03 | (0.01) | <0.001 |
| Male | −0.06 | (0.41) | 0.887 | 0.84 | (0.36) | 0.018 | −0.33 | (0.44) | 0.450 |
| College educated | −0.97 | (0.34) | 0.004 | −0.04 | (0.30) | 0.882 | 0.41 | (0.34) | 0.222 |
| Partnered | −0.02 | (0.34) | 0.947 | −0.23 | (0.30) | 0.443 | 0.02 | (0.34) | 0.956 |
| Number of diagnoses | 0.37 | (0.16) | 0.019 | 0.18 | (0.13) | 0.190 | 0.61 | (0.15) | <0.001 |
| History (years) | −0.02 | (0.01) | 0.164 | −0.04 | (0.01) | 0.004 | −0.01 | (0.01) | 0.643 |
| Male × Time | 0.01 | (0.04) | 0.852 | −0.05 | (0.04) | 0.163 | −0.03 | (0.04) | 0.366 |
| College × Time | 0.03 | (0.03) | 0.328 | −0.01 | (0.03) | 0.655 | −0.02 | (0.03) | 0.424 |
| Partnered × Time | −0.04 | (0.03) | 0.240 | 0.01 | (0.03) | 0.648 | 0.01 | (0.03) | 0.688 |
| Number of diagnoses × Time | 0.03 | (0.01) | 0.062 | 0.02 | (0.01) | 0.174 | −0.01 | (0.01) | 0.300 |
| History (years) × Time | 0.00 | (0.00) | 0.267 | 0.00 | (0.00) | 0.397 | 0.00 | (0.00) | 0.776 |
| Intercept | 10.19 | (0.57) | <0.001 | 10.93 | (0.48) | <0.001 | 30.98 | (0.48) | <0.001 |
| Random effect | | | | | | | | | |
| SD (Time) | 0.15 | (0.03) | | 0.14 | (0.02) | | 0.14 | (0.02) | |
| SD (Intercept) | 20.83 | (0.12) | | 20.26 | (0.11) | | 30.14 | (0.11) | |
| SD (Residual) | 20.64 | (0.08) | | 20.45 | (0.08) | | 20.17 | (0.08) | |

Notes:
1. Quadratic growth curve fit to data using full-information maximum likelihood (FIML) estimation.
2. Random effects indicate the variability in the fixed effect. For example, the 95% confidence interval for the slope (Time) of Anxiety is (from −0.53 to 0.05) = (Fixed-effect estimate for Time) ± (Random-effect SD for Time × 1.96) = −0.24 ± (0.15 × 1.96).
3. Interaction terms with time indicate change in outcome for the time-invariant factor (i.e., male, college, partnered, number of diagnoses, and disease history) over time.

period while the other changed noticeably within the first 6 months of follow-up, and then it reversed course back toward the baseline value (Fig. 1). That is, on each outcome some participants were in a decay-of-impact group and the others were not. About half of the participants were in the decay-of-impact group: anxiety 45.6%, depression 50.7%, and communication with physicians 46.3%.

Participants who had decay of impact on one of the two mental-health outcomes (anxiety or depression) were also likely to be classified as having decay of impact on the other one (Phi = 0.508, Appendix 3). However, participants who had decay of impact on one of the two mental-health outcomes were no more or less likely to have decay of impact on communication (Phi = 0.095 and 0.043, Appendix 3).

On the two mental-health outcomes, the decay-of-impact group was the group with worse baseline status: more symptoms, and more-frequent symptoms, of anxiety and depression. In contrast, on communication with physicians the decay-of-impact group was the group with better baseline status: more frequent use of the three specified methods for good patient-physician communication. Also, by the end of the follow-up year the anxiety and depression scores had decayed back to their respective baseline levels,
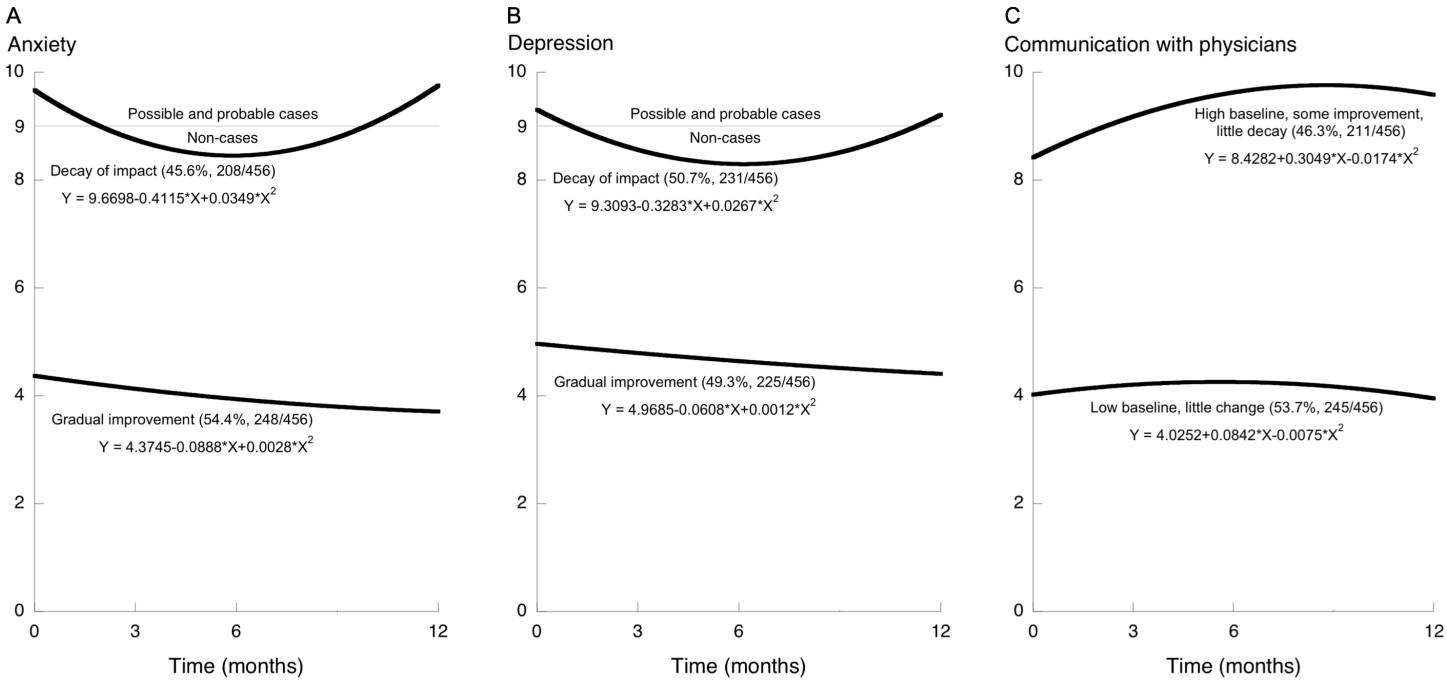

**Figure 1 Trajectories of change after health education, showing two trajectory-defined groups for each of the three outcomes.** Growth-mixture modeling revealed two trajectory-defined groups for each outcome. On anxiety and depression higher scores are worse (A and B). On communication with physicians higher scores are better (C). For each outcome, one of those two groups had improvement followed by deterioration: decay of impact. For anxiety and depression, a score of 9 is the cutoff used in Japan to separate non-cases from possible and probable cases.

whereas on communication the decay trajectory was clear but the scores did not return to the baseline level, in other words the decay itself was smaller on communication with physicians than on the mental-health outcomes (Fig. 1).

## Contributors to group membership (Table 3)

For all three outcomes, self-efficacy at baseline was associated with group membership. Participants with higher self-efficacy were more likely to be in the group with lower anxiety at baseline, in the group with lower depression at baseline, and in the group with better communication at baseline.

There were no noteworthy associations between group membership and gender, civil status (i.e., partnered or not), or the number of years of chronic-disease history. College education was associated with group membership on only one of the three outcomes (anxiety).

For anxiety and for communication with physicians, the number of diagnoses was also associated with group membership. Regarding anxiety, participants with more diagnoses were more likely to be in the group with higher (i.e., worse) scores at baseline and subsequent decay of impact. Regarding communication, participants with more diagnoses were more likely to be in the group with higher (i.e., *better*) scores at baseline and subsequent decay of impact.

**Table 3 Factors predicting membership in groups defined by their trajectory after the CDSMP.**

| Factor | Anxiety | | | Depression | | | Communication | | |
|---|---|---|---|---|---|---|---|---|---|
| | Adjusted odds ratio | Std err | p-value | Adjusted odds ratio | Std err | p-value | Adjusted odds ratio | Std err | p-value |
| Male | 0.92 | (0.23) | 0.730 | 1.50 | (0.38) | 0.112 | 0.62 | (0.15) | 0.050 |
| College | 0.59 | (0.12) | 0.009 | 0.83 | (0.17) | 0.376 | 1.17 | (0.23) | 0.432 |
| Partnered | 0.99 | (0.20) | 0.947 | 0.78 | (0.16) | 0.227 | 1.17 | (0.23) | 0.418 |
| Self-efficacy | 0.97 | (0.01) | <0.001 | 0.95 | (0.01) | <0.001 | 1.02 | (0.01) | 0.013 |
| Number of diagnoses | 1.23 | (0.12) | 0.032 | 1.12 | (0.11) | 0.257 | 1.30 | (0.13) | 0.006 |
| History (years) | 0.99 | (0.01) | 0.266 | 0.99 | (0.01) | 0.244 | 1.00 | (0.01) | 0.683 |
| Intercept | 2.66 | (0.99) | 0.009 | 6.37 | (2.49) | <0.001 | 0.29 | (0.11) | 0.001 |

Notes:
1. Values in parenthesis are standard errors.
2. The results shown are from logistic regression. The 0–1 coding of group membership (which is the dependent variable) reflects the relative magnitudes of the baseline scores. For all outcomes, the group with the lower baseline score is coded "0" and the group with the higher baseline score is coded "1." Thus, the group with less anxiety at baseline is coded "0" while the group with more anxiety at baseline is coded "1." The same is true for depression. In contrast, the group with better communication (higher scores) at baseline is coded "1" and the group with worse communication (lower scores) at baseline is coded "0."

# DISCUSSION

## All participants

When all participants were considered together, all three outcomes improved over the first 6 months. That improvement was followed by a small deterioration. Thus, even from the least-detailed analyses, some decay of impact was evident (Table 1). That interpretation is supported by the results of the growth-curve analyses: change over time was curvilinear (quadratic) for all three outcomes (Table 2).

The growth-curve analyses also showed that higher self-efficacy at baseline was associated with less anxiety, less depression, and better communication with physicians, which is consistent with the theoretical basis of the CDSMP (*Lorig & González, 1992*).

Also evident at this level of analysis were associations with multimorbidity. Having more diagnoses was associated with more anxiety, more depression, and *better* communication with physicians. Of those three findings, the first two might be expected, but the third is particularly interesting. It is also reflected in the analyses of membership in trajectory-defined groups, and so we will discuss it below.

## Findings from GMM: groups defined by their trajectories

For all three outcomes the results of GMM were consistent: Among all of the models tested, the two-group models had the lowest BIC and the highest $P_c$. We are therefore confident in saying that GMM revealed *two* latent groups among these participants in the CDSMP. In some circumstances practical considerations could override the conclusions from those statistical criteria, as described in Appendix 2.

As noted above in the Introduction, in some previous studies, subsets of CDSMP participants were defined a priori and with reference to theory. In contrast, groups identified by GMM are empirical, as is each participant's group membership. We note that the GMM approach can lead to testable hypotheses (regarding multimorbidity, as

described below), and it can be used to answer important questions about whether similar phenomena also occur among other groups and in different settings.

## Factors affecting self-management

These findings are not inconsistent with the five categories of factors affecting self-management that were identified by *Schulman-Green et al. (2016)*. It is noteworthy that the CDSMP does address factors in all five of Schulman-Green's categories: self-efficacy (which is in the category of "personal/lifestyle characteristics"), problem-solving to cope with symptoms (in the category of "health status"), social support via the CDSMP group sessions (in the category of "resources"), action plans for ensuring proper nutrition (in the category of "environmental characteristics"), and proactive patient-physician communication (in the category of "health care system").

With regard to the decay of impact, we note that Table 3 shows associations of trajectory-group membership with two factors that were identified by *Schulman-Green et al. (2016)* as influencers of self-management. Those two factors are self-efficacy (in their "personal/lifestyle characteristics" category) and the number of diagnoses (i.e., comorbidity, in their "health status" category). In addition to those two, the baseline level of patient-physician communication is in Schulman-Green's category of "health care system." It is quite possible that factors in Schulman-Green's two other categories ("resources" and "environmental characteristics") also affect the decay of impact, and we believe this is an important topic for future research.

## Mental health and reinforcement

For both mental-health outcomes, the trajectory-defined groups differed in their baseline status and in their pattern of change after the program. Regarding anxiety, approximately half of the participants were in a group that began with relatively good scores, and they improved very gradually over the following year. In contrast, the other half were in a group that began from a high-anxiety baseline. That second group improved over the first 6 months, but by the time of the 12-month follow-up it had returned to its baseline level, and thus we refer to the latter group as the decay-of-impact group. The same was true with regard to depression.

Dichotomization is undoubtedly dangerous (*Harrell, 2019*), and yet HADS scores are used to separate people into categories of anxiety and depression severity. In Japan, the HADS threshold score separating non-cases from possible and probable cases was 9 (*Matsudaira et al., 2009*). The decay-of-impact trajectories on both anxiety and depression crossed that threshold twice—first during the improvement occurring soon after the intervention ended, and then again about 8 months later during the decay back toward the baseline value (Figs. 1A and 1B). Therefore, to the extent that the threshold of 9 is useful, both the improvement measured soon after the CDSMP and also the deterioration measured near the end of the follow-up year were clinically important.

"Average therapeutic trial results can mislead" (*Moynihan, Henry & Moons, 2014*), but GMM provides more detail than average results. Here GMM showed that only some of the participants had decay of impact. At least on anxiety and depression, both the existence

of a decay-of-impact group and the movement of that group between clinical categories support the idea that follow-up interventions—reinforcement—should be offered to *some* of the participants. Had reinforcement been given to all, it is unlikely that those in the group without decay of impact would have benefitted from it, simply because they already had almost no psychological distress—almost no room to improve. Rather than being expended on all of the participants, the resources used to implement reinforcement should be saved for the people who need it, to help them maintain their newly-improved status or perhaps improve further.

The present findings show how the small effect sizes and null results in published studies of reinforcement could be underestimates. Specifically, when all participants are considered together (Table 1), then any benefits to a group with decay-of-impact (Fig. 1) will become at least more difficult to detect, and possibly even completely effaced, as they are mixed with the much smaller (or even zero) benefits to the non-decay group. This dilution is clearly shown in the difference between Table 1 and Fig. 1. Table 1 shows only a very small decay of impact, because all participants were considered together, but Fig. 1 shows substantial decay of impact because GMM separated the participants into two trajectory-defined groups as it "unmixed" the data. If one were to look at Table 1 alone, one could easily conclude that the benefits of the program and also the decay of impact (i.e., the need for reinforcement) were both very small. But that would be another example of how average results can mislead (*Moynihan, Henry & Moons, 2014*). Figure 1 shows that almost half of the participants had both important benefits and important decay of impact, and the difference between Table 1 (the diluted results) and Fig. 1 (the post-GMM results) shows how mixing the data from two different groups caused important effects to be diluted.

Testing the effects of reinforcement is reasonable, but only in those whom reinforcement could benefit: the decay-of-impact group. To identify that group at baseline, that is, even before the CDSMP begins, the present findings offer two potential criteria: a high baseline score (greater distress) on the mental-health outcome of interest, and a low level of self-efficacy. With regard to anxiety, a third criterion could be multimorbidity.

### Communication with physicians

Similar to the results described above for mental health, regarding communication with physicians GMM revealed two groups, each comprising about half of the participants, and those two groups began from noticeably different baselines. One group ($n = 245$) started from a very low baseline communication score (about 4 points) and it changed very little over the following year (Fig. 1). This could well indicate an unmet need. Specifically, by the standard implied in the patient-physician communication scale, for those 245 participants substantial improvement after the baseline measurement was possible, but it did not occur. This leads to at least three research questions: (1) Are some participants in fact satisfied with a "low" level of communication? (2) Was the program implemented as well as possible? (3) Even if the implementation was good, would those 245 participants have benefitted from a more-intensive intervention with an even-greater emphasis on practicing communication skills?

Also noteworthy are the three criteria that might be used to pre-emptively identify the participants who are most likely to need communication-skill practice: a low communication score at baseline, low self-efficacy, and *uni*morbidity.

## Multimorbidity

The participants with more diagnoses had better communication scores in the initial growth-curve analysis (Table 2), and they were more likely to be in the trajectory-defined group that had better communication scores throughout the year (Table 3). While the CDSMP has been found to be particularly useful to people with multiple diagnoses (*Harrison et al., 2012*), here multimorbidity was associated with a desirable health-related behavior even at baseline. To address the apparent connection between having multiple diagnoses and communicating well with physicians, we begin by noting that the communication scores reflect how often the respondents do the following three activities: making a list of questions to ask one's physician during clinic visits; asking one's physician about things that one wants to know or does not understand regarding one's treatment; and discussing (with one's physician) personal problems related to one's medical condition. The people with multiple diagnoses probably had more experience being in health-related situations that were difficult to manage. To deal with those difficulties, perhaps they began writing lists of questions, asking for clarification, and discussing personal issues related to their diseases, simply because their health conditions were so complex. We hypothesize that at least some of the people with multimorbidity had become accustomed to doing those activities, and therefore in the domain of patient-physician communication they had already become "expert patients" (*Reeves et al., 2008*) by the time the study began. To the extent that better patient-physician communication results in better clinical care, this connection between multimorbidity and good communication could account at least in part for the documented association of multimorbidity with higher-quality care (*Min et al., 2007*).

Other factors could also be important. For example, self-selection might have played a role. After all, participation in the program was voluntary (as it is worldwide). Personality can moderate the effects of the CDSMP (*Franks et al., 2009*; *Jerant et al., 2010*), and perhaps it also affects one's decision to participate. Among all of the eligible people with multimorbidity, those who are less "conscientious" and less interested in self-managing their conditions would not often make lists of questions, etc., and they might not have found the CDSMP to be attractive and thus would not have participated. In contrast, the CDSMP might appeal to people like the highly-proactive communicator with eight chronic conditions who was described by *Haslam (2015)*. People with multiple diagnoses who take initiative in self-managing their condition(s) by writing lists of questions, etc. could be over-represented among the program's participants.

As explanations of the association between multimorbidity and good patient-physician communication, both the self-selection hypothesis and the already-an-expert-patient hypothesis remain to be tested, and of course they are not mutually exclusive: both could be true.

## Generalizability

Because this study involved not a sample but a census of the population of interest, the common concerns about generalizability are not strictly applicable. Chronic-disease self-management interventions have been implemented in many countries, but the country-specific differences in those implementations are important (*O'Connell, McCarthy & Savage, 2018*), and we do not claim that the results shown in Fig. 1, Table 3, etc. necessarily apply to CDSMP participants outside Japan. Nonetheless, we do suggest that the approach and the methods employed here could be quite useful in evaluations of the CDSMP and of other health-education programs worldwide. As a minimum, evidence explaining why effect sizes appear to be small is important wherever the CDSMP is used. In addition, a generalized application of GMM and similar methods would ground the understanding of these programs' effects more firmly in the empirical reality of trajectory-defined groups. Related analytic methods may also be useful, including methods that allow analyses of individual participants' trajectories (*Kozlowski et al., 2013*). A generalized application of the approach and the methods used in the present study would also help to meet the current need for research on long-term maintenance of self-management skills (*Miller et al., 2015*).

## Cost-effectiveness

Cost-effectiveness analyses of the CDSMP indicate that it results in net savings (*Ahn et al., 2013*). Its cost-effectiveness has also been demonstrated through analysis of the incremental cost-effectiveness ratio per quality-adjusted life year (*Basu et al., 2015*). Formal cost-effectiveness analyses of the prediction of CDSMP trajectory-group membership are beyond the scope of the present study. Nonetheless, it is reasonable to expect the costs to be low. Specifically, for the three outcomes of interest here, the present results indicate that predicting trajectory-group membership might require completion of only the 14-item HADS, the 6-item self-efficacy scale, the 3-item patient-physician communication scale, and a question-item regarding multimorbidity— all at baseline only. The participants already fill out registration forms before the start of the first CDSMP group session. If the burden on the participants must be strictly minimized, then the necessary information could be gathered only once (at baseline), and it would take only a few extra minutes.

While the costs would be expected to be low, the effectiveness (benefits) could be high, because there is potential for prevention or minimization of the decay of impact. Specifically, as shown in Table 3 and in Fig. 1, baseline information might be used to predict who among the participants will have decay of impact, and thus who is most likely to benefit from extra attention during the program and from reinforcement after the program. This could strengthen the empirical foundation for decisions about cost-effective allocation of those educational resources.

## Limitations

The four waves of data collection over 1 year were more than enough to allow detection of decay of impact, but more frequent measurement and longer follow-up would of course be useful.

The number of diagnoses was self-reported. While we would have preferred to use medical records, for many chronic conditions self-reported diagnosis is accurate enough for research (*Karison et al., 1999*; *Wada et al., 2009*). The diagnosis of clinical depression was found by *Sanchez-Villegas et al. (2008)* to be over-reported, with a true-positive percentage of 74.2%. In that study, among 62 people who self-reported a diagnosis of depression, 46 were found to be true positives according to the Structured Clinical Interview for DSM-IV (SCID-I) (46/62 = 0.742). It is possible that in the present study a similar percentage of those who self-reported that they had been given a diagnosis of depression would in fact have met the SCID-I criteria for depression. In that case, the number of true positives would be estimated to be approximately 19 rather than the 26 who did self-report that they had been given a diagnosis of depression (Appendix Table A1C): 0.742 × 26 = 19.3. This is unlikely to have had a large effect on the main results or conclusions. The logistic regression analysis did identify the number of diagnoses as a predictor of trajectory-group membership (Table 3), but any effect of over-reporting of depression on that result was probably very small. Specifically, 26 − 19 = 7, and 7/456 = 0.015. Thus, self-reporting of a diagnosis of depression might have caused the number of participants with depression to have been over-estimated by approximately 7, which is 1.5% of the total. In addition, it is important to remember that the number of diagnoses was not included in the GMM analysis, so any inaccuracy or imprecision in that number did not affect the GMM results (Fig. 1 and Appendix 2). Also, the presence or absence of any self-report of any specific diagnosis (depression, etc.) was not included in the GMM analysis, so any inaccuracy or imprecision in those data also did not affect the GMM results.

## Previous knowledge gaps, new knowledge, and applications

One approach to the presentation of research is to explicate knowledge gaps, how those gaps have been "filled," and the practical consequences thereof. With that in mind, we give such information below.

Knowledge gap 1: The benefits of the CDSMP appear to be small (and reinforcement appears to be ineffective), but the reason is not well understood. Filling knowledge gap 1, we found how mixing of data from two distinct trajectory-defined groups can make the overall benefits of the CDSMP appear to be small even though they are relatively large for some participants. Also, one of those two trajectory-defined groups was characterized by decay of impact. Given the new information obtained in filling that first gap, another knowledge gap became clear.

Knowledge gap 2: Criteria for identifying people who are likely to have decay of impact are not known. Filling knowledge gap 2, we found that the people who are most likely to have decay of impact can be identified using baseline self-efficacy and the number of diagnoses (Table 3). Also, people who eventually had decay of impact were those with higher baseline scores on the outcome of interest (Fig. 1).

Applications of new knowledge: This new knowledge can be useful in at least two ways. First, it can be used to tailor reinforcement to the participants who are most likely to need reinforcement, which could increase cost-effectiveness. Second, it can be used to

establish two new goals for CDSMP implementation and two new criteria for CDSMP evaluation: a low prevalence of decay of impact, and a small magnitude of decay of impact (*Park et al., 2012*).

### Considerations regarding theory

As noted above, the theoretical foundation of the CDSMP is centered on the concept of self-efficacy (*Bandura, 2019*). In light of the present results, other theories and conceptual frameworks may also be useful, especially those emphasizing long-term outcomes (*Miller et al., 2015*). For example, the field of relapse prevention after treatment for substance abuse has well-studied theories and practices (*Hendershot et al., 2011*; *Menon & Kandasamy, 2018*; *Witkiewitz & Marlatt, 2004*), some of which might be adapted to inform strategies for preventing decay of impact after chronic-disease self-management education. In addition, consideration should be given to the difference between *changing* health-related behavior and *maintaining a new* health-related behavior for long-term benefits (*Rothman, 2000*; *Sciamanna et al., 2011*; *Joseph et al., 2016*; *Giacobbi, 2016*).

## CONCLUSIONS

Growth-mixture modeling exposed two trajectory-defined groups, and the CDSMP clearly benefitted one group more than the other. However, the group that benefitted also had substantial decay of impact, and thus needed reinforcement. The decay-of-impact group comprised almost half of the participants. At baseline (i.e., before the program began), the participants most likely to need reinforcement were those with multimorbidity, those with low self-efficacy, and those who were clinically anxious or depressed.

Once the participants who are likely to have decay of impact are identified, extra attention and reinforcement can then be tailored. They can be focused specifically to benefit the people with the greatest need.

## ACKNOWLEDGEMENTS

The authors would like to express their gratitude to the Japan Chronic Disease Self-Management Association, as well as to all of the people who participated in the study. MJ Park is grateful for advice received from Y Yamazaki, for help and collaboration provided by the self-management research team at the University of Tokyo, for administrative assistance and technical support received from N Okamoto, and for advice and academic support received from T Kiuchi and H Ishikawa.

### Funding

This work was supported by Japan's Ministry of Health, Labor, and Welfare aids for Scientific Research. No additional external funding was received for this study. The funders had no role in study design, data collection and analysis, decision to publish, or preparation of the manuscript.

## Grant Disclosures

The following grant information was disclosed by the authors:

Japan's Ministry of Health, Labor, and Welfare aids for Scientific Research.

## Competing Interests

The authors declare that they have no competing interests.

## Author Contributions

- Min Jeong Park conceived and designed the experiments, performed the experiments, analyzed the data, contributed reagents/materials/analysis tools, authored or reviewed drafts of the paper, approved the final draft, and was very directly involved in collecting the data: distributing questionnaires, receiving completed questionnaires, sending reminders when questionnaires were not returned on time, speaking directly with participants to answer their questions about the study, etc.
- Joseph Green conceived and designed the experiments, performed the experiments, analyzed the data, contributed reagents/materials/analysis tools, prepared figures and/or tables, authored or reviewed drafts of the paper, approved the final draft.
- Hun Sik Jung conceived and designed the experiments, performed the experiments, analyzed the data, contributed reagents/materials/analysis tools, authored or reviewed drafts of the paper, approved the final draft.
- Yoon Soo Park conceived and designed the experiments, performed the experiments, analyzed the data, contributed reagents/materials/analysis tools, prepared figures and/or tables, authored or reviewed drafts of the paper, approved the final draft.

## Human Ethics

The following information was supplied relating to ethical approvals (i.e., approving body and any reference numbers):

This study was approved by the Research Ethics Committee of the Graduate School of Medicine at the University of Tokyo (number 1472 - (2)).

## Data Availability

The raw data are available in the University of Tokyo's Repository: http://hdl.handle.net/2261/00077116.

## Supplemental Information

Supplemental information for this article can be found online at http://dx.doi.org/10.7717/peerj.7229#supplemental-information.

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
