# Peer review of "Trajectories of change after a health-education program in Japan: decay of impact in anxiety, depression, and patient-physician communication"

_PeerJ, doi:10.7717/peerj.7229_

## Round 0.1 · original submission · Major Revisions

This is an interesting article on trajectories of change after a health-education program. Although it has merits, its current form still has some issues, which need to be addressed before publication. Please consider these review comments and revise it accordingly.

Reviewer 1 ·

Basic reporting

Background: In the title as well as the literature review, it is not clear whether the term trajectory of change refers to (1) a post-intervention change in behavior , (2) a change in the course of chronic illness as a result of the change in behavior brought about by health education, or (3) the change in sustained application of learned skills in specific chronic disease self-management?
. The authors need to clearly articulate what they mean by trajectory of change. The context of what is being discussed needs a little more expansion in the introduction, especially the concept of decay of impact after health education.

Experimental design

The authors need to address or re-articulate clearly the following questions.
1.What is the study population?
2.What is the sampling method used?
What is/are the research question(s)? What is/are the dependent variable (s)or the outcome(s)? And what are the independent variables? The research question should be well defined, relevant & meaningful.

Validity of the findings

The authors need to address or re-articulate clearly the answers to the following questions, which impact the validity of their findings.
What is the established validity and reliability of each of the research instruments?
Factors affecting the outcome of a chronic disease self-management program have been identified in a prior research (Schulman‐Green, et al., 2016). Accordingly, they include "personal/ lifestyle characteristics, health status, resources, environmental characteristics and health care system". It appears that health education is just one piece in a large puzzle and may not appear to explain all the reasons for decay of impact. The authors need to address all these and other confounding factors.
The authors mention that they resorted to self-reported diagnosis of chronic conditions instead of medical records because self-reported diagnosis is accurate enough for research. Nonetheless, this reviewer would argue that self-reported diagnosis is exactly what it is: self-reported; and everyone who feels “bad, sad, or mad” may report they are anxious or depressed As a matter of fact, the validity of a self-reported diagnosis of depression among participants in a cohort study even using Structured Clinical Interview, which is the Gold Standard, was found to be 74.2%, Sanches-Villegas et al., (2008). Although audience analysis may be one of the important first steps to accomplish before conducting a health education program, the benefits of conducting a survey of the type suggested by the authors in this study every time a health education is planned should be weighed against the cost-effectiveness of such an intervention.
References
- Schulman‐Green, D., Jaser, S. S., Park, C., & Whittemore, R. (2016). A metasynthesis of factors affecting self‐management of chronic illness. Journal of advanced nursing, 72(7), 1469-1489.
- Sanchez-Villegas, A., Schlatter, J., Ortuno, F., Lahortiga, F., Pla, J., Benito, S., & Martinez-Gonzalez, M. A. (2008). Validity of a self-reported diagnosis of depression among participants in a cohort study using the Structured Clinical Interview for DSM-IV (SCID-I). BMC psychiatry, 8(1), 43.

Additional comments

In sum, the authors need to state how the research fills an identified knowledge gap and how the new knowledge will be used to influence population health policies and programs in the application of health education in chronic disease self-management programs in the study area and beyond.

Reviewer 2 ·

Basic reporting

I commend the authors for the narration of the problem right from the introductory part of the manuscript. Nonetheless, many of the literature cited is rather too old; it is suggested that more recent literature is added to this section.

Experimental design

1. It seems that the study data are solely based on self-reported responses acquired from the respondents therefore, it is essential for the reliability of such responses to be carried out prior to any further analysis. No, any reliability analysis is reported within the manuscript.
2. A more detail explanation on how the GMM is implemented needs to be highlighted in order to allow replication of the analysis in other studies.

Validity of the findings

1. More explanation should be provided within the groups defined by trajectories supported by the literature
2. Recent literature should be cited to support the findings of the study. Many of the literature cited is quite old.

Additional comments

The authors assessed the effectiveness of a health education programme in benefiting patients with chronic diseases. It was inferred that the benefit of the Chronic Disease Self-Management Program in Japan could be effectively measured through the employment of GMM as well as the Logistic regression analysis with respect to more sample size. Overall, the manuscript is well written, and the findings might be useful to the health educators and other relevant stakeholders in evaluating the effectiveness of health intervention programmes. However, a number of issues need to be addressed to improve the quality of the manuscript.

Reviewer 3 ·

Basic reporting

This paper is useful for studying health education and behavioral science. The researcher has presented his writing well along with the necessary descriptions. However, detailed reporting of the research process needs to be added.
Elaboration related to the behavioral theory discipline needs to be given to provide an understanding of the operation of the theory of behavior for readers.

Experimental design

No comment

Validity of the findings

Researchers need to present a more detailed explanation of the guarantee of validity in sample selection, data collection and data analysis

Additional comments

this paper is useful for the development of behavioral science especially health education. Researchers need to present more detailed research processes so that other researchers can use it to conduct similar research. In general, the use of analytical methods (GMM) needs to bring up aspects of benefits in their use in health education applications in the field.
. I give some suggestions to the author
1. Demographic and clinical information were collected using self-administered (line 114) . Author need to add such explanation how to guarantee the precision of information collected.
2. In the research background the author mentioned this statement: Health education can benefit people with chronic diseases. However, in previous research those benefits were small, and reinforcement to maintain them was not effective. A possible explanation is that the benefits appeared to be small and reinforcement appeared to be ineffective because those analyses mixed data from two latent groups: one group of people who needed reinforcement and one group of people who did not. The hypothesis is that mixing the data from those two different groups caused the true effects to be “diluted”. I suggest author to add some elaboration in discussion section to benefit your finding pertaining with this statement.
3. To test that hypothesis we used data from the Chronic Disease Self-Management Program in Japan, focusing on anxiety, depression, and patient-physician communication. Author need to explain why that type of disease was chosen instead of others.
4. Data were collected from participants in the CDSMP in Japan. They were recruited through public service centers and the Internet homepage of the Japan Chronic Disease Self Management Association (Japan Chronic Disease Self-Management Association, 2018). (line 101-103). On my point of view author need to explain which sampling method applied to this research a little bit more detail?
5. Data were collected four times over one year. Author need to explain how to prevent bias? By repeating the same question, respondent may learn and create certain respond.
6. How about the participation rate? Did all of them submit respond? Please give some explanation.
7. In result section, from 456 participants were analyzed. Among them, 79% were women, 48% were college educated,52% of respondents were partnered (married or living with someone) (line 155). In my view, the author needs to elaborate about the differences in research results between groups of respondents, given that the dynamics of information sharing between groups may be different.

---

## Round 0.2 · accepted · Accept

Thank you for your revisions.

Reviewer 1 ·

Basic reporting

This reviewer commends the authors for adequately addressing the questions, comments and clarifications requested previously on basic reporting.

Experimental design

The questions and comments made on research methodology, including research questions and method of data collection have been well addressed.

Validity of the findings

The authors have provided the validity and reliability of survey instruments used in their research as requested. The authors have provided point by point response to the reviewer's questions and have incorporated most of the comments made by this reviewer.

Additional comments

I commend the authors' for their attention to detail in their responses to questions, comments, and requests for clarification.

Reviewer 2 ·

Basic reporting

It is sufficient

Experimental design

It has been revised as suggested

Validity of the findings

Adequately addressed

Additional comments

I thank the authors for revising the manuscript as per suggested.